# Selection of isomerization pathways of multistep photoswitches by chalcogen bonding

Shuaipeng Jia[1,2], Hebo Ye[1], Peng He[1], Xin Lin[1,2] & Lei You[1,2,3]

Multistep photoswitches are able to engage in different photoisomerization pathways and are challenging to control. Here we demonstrate a multistep sequence of E/Z isomerization and photocyclization/cycloreversion of photoswitches via manipulating the strength and mechanism of noncovalent chalcogen bonding interactions. The incorporation of chalcogens and the formyl group on open ethene bridged dithienylethenes offers a versatile skeleton for single photochromic molecules. While bidirectional E/Z photoswitching is dominated by neutral tellurium arising from enhanced resonance-assisted chalcogen bonding, the creation of cationic telluronium enables the realization of photocyclization/cycloreversion. The reversible nucleophilic substitution reactions further allow interconversion between neutral tellurium and cationic telluronium and selection of photoisomerization mechanisms on purpose. By leveraging unique photoswitching patterns and dynamic covalent reactivity, light and pH stimuli-responsive multistate rewritable materials were constructed, triggered by an activating reagent for additional control. The results should provide ample opportunities to molecular recognition, intelligent switches, information encryption, and smart materials.

Photochromic switches are among the most sought-after molecular switches because of light-responsive reversible interconversion, high spatiotemporal resolution, and broad applications in many aspects[1–9]. It is thus of high significance to develop new structures and regulating mechanisms of photoswitches. Configurational E/Z isomerization and electrocyclization are two main photoisomerization mechanisms[10–16]. Dithienylethenes (DTEs) adopting a cyclic covalent ethene bridge represent one popular class of photoswitches and proceed via photocyclization/cycloreversion (Z/C)[17–20]. In contrast, open-bridged DTEs are able to participate in multistep E/Z and Z/C pathways (Fig. 1a), rendering them difficult targets to control[21–23]. Moreover, open-bridged DTEs would have the advantage of diverse modification and functionalization on the ethene bridge. To replace covalent linkages for constrained structural flexibility we were wondering the possibility of noncovalent interaction bridged photoswitches. Despite challenging the modulation of different photoisomerization pathways on demand would offer opportunities for multistage switching and multistate responses[24–29], enhancing the complexity and enriching information encoding.

Chalcogen bonding as one type of emerging supramolecular bonding forces is generating intensive interest, which can be utilized in in guest binding and transport, crystal engineering, homogeneous catalysis, and photovoltaic materials[30–36]. Both electrostatics and orbital delocalization play a stabilizing role in chalcogen bonding, showing strong directionality[37,38]. The structural bias induced by chalcogen bonds was recently explored to harness azobenzene and hemithioindigo E/Z photoswitches[39–42]. The regulation of multistep photoswitching with chalcogen bonds is untouched, to the best of our knowledge. By placing thioether/selenoether/telluroether and a formyl group on vicinal positions of DTEs (Fig. 1b), we postulated that the

[1]State Key Laboratory of Structural Chemistry, Fujian Institute of Research on the Structure of Matter, Chinese Academy of Sciences, Fuzhou 350002, China. [2]University of Chinese Academy of Sciences, Beijing 100049, China. [3]Fujian Science & Technology Innovation Laboratory for Optoelectronic Information of China, Fuzhou 350108, China. ✉e-mail: lyou@fjirsm.ac.cn

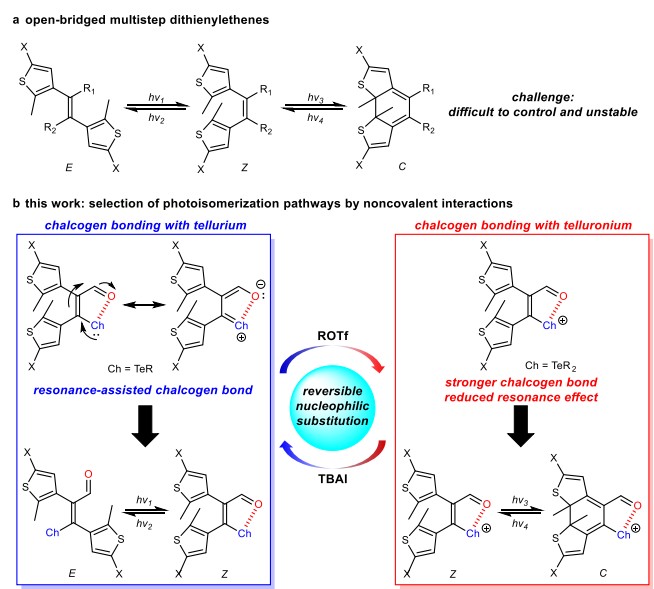

**Fig. 1 | Molecular design of open-bridged multistep dithienylethenes and the working mechanism. a** *E/Z* photoisomerization and photocyclization/cycloreversion (*Z/C*) for open-bridged dithienylethene multistep photoswitches. Abbreviation *C* (closed) was used to label photocyclization product. **b** This work of selection of photoisomerization pathways in chalcogen bond-containing dithienylethene photoswitches, controlled by chalcogen bonds and reversible nucleophilic substitution reactions (Ch highlighted in blue, O highlighted in red). TBAI tetra-*n*-butylammonium iodide.

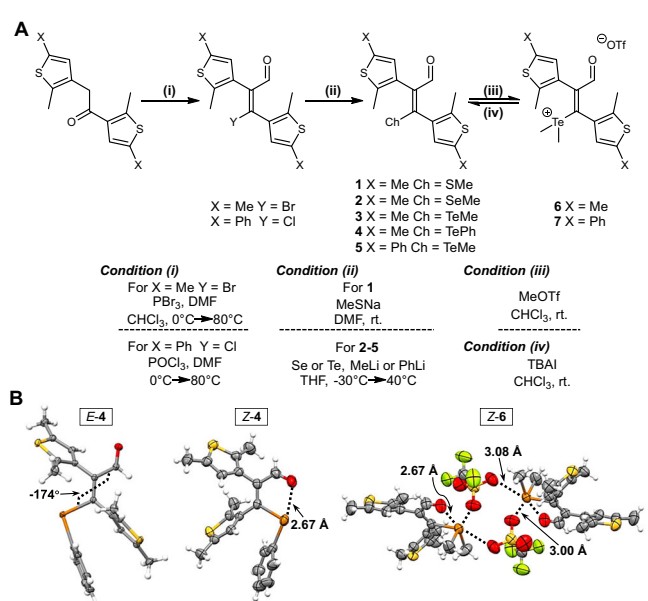

**Fig. 2 | Modular synthetic pathway and crystal structures. A** Synthesis of chalcogen bond-containing dithienylethene photoswitches (*E* isomer is shown).
**B** Crystal structures of *E*-**4**, *Z*-**4**, and *Z*-**6**, with the distances of chalcogen bonding listed. Displacement ellipsoids are drawn at the 50% probability level, with hydrogen atoms as spheres of arbitrary radii.

adjacent chalcogen bond (Ch···O) would regulate thermodynamic stability and photoisomerization of *E/Z* isomers. Furthermore, with resonance-assisted chalcogen bonding[43,44] delocalizing the central double bond in *Z* configuration the 6π-electrocyclization of hexatriene motif to afford closed-ring isomer (*C*) would be impacted. Moreover, the formation of cationic telluronium would strengthen chalcogen bonding and yet potentially sabotage resonance effect[45,46]. Therefore, the manipulation of chalcogen bonds would provide a promising platform for selecting from *E/Z* isomerization and photocyclization/cycloreversion (*Z/C*) in a single photochrome system.

In the current work the chalcogen bonding within open-bridged DTEs afforded a versatile avenue toward controlling multistep photoswitching of *E/Z* isomerization and photocyclization/cycloreversion (Fig. 1b). Bidirectional *E/Z* configurational photoswitching dominated for telluroether due to enhanced resonance-assisted chalcogen bonding. The creation of telluronium allowed the realization of photochemical ring-closure/ring-opening (*Z/C*), with switching patterns of neutral tellurium and cationic telluronium further selected by reversible nucleophilic substitution reactions on purpose. Finally, to demonstrate the potential photoaddressed switchable rewritable materials were created. This work adds into the toolbox of multistep photoswitches, with structural and mechanistic insights setting the stage for future design and functionalization studies.

## Results and discussion
### Synthesis and structures
To realize the strategy, a modular synthetic pathway was designed (Fig. 2A and Supplementary Figs. 1–54). The key intermediate 1-bromo/chloro-2-formyl-1,2-di(2-methyl-5-X-3-thienyl)ethene was obtained via Vilsmeier-Haack-Arnold formylation type reactions[47]. The following nucleophilic substitution afforded the desired photoswitches (**1**–**5**). To probe substituent effect three neutral tellurium derivatives were prepared (**3**–**5**). While the *E* isomer was isolated for **1** and **2**, both *E* and *Z* isomers could be attained for their Te analogs. Moreover, the

individual reactions of **3** and **5** with methyl triflate gave their telluronium salts (**6** and **7**).

X-ray crystal analysis was then conducted to offer structural insights. The near coplanarity of Te atom, formyl group, and olefin was found for *E*-**4**, as also the case for crystal structures of *E*-**2** and *E*-**3** (Fig. 2B and Supplementary Fig. 55). *Z*-**4** adopts an intramolecular chalcogen bond and has a Te···O distance of 2.67 Å (Fig. 2B). Furthermore, the residence of aldehyde oxygen along the extension of Te-C bond supports the directionality of chalcogen bond. Chalcogen bonding between TeMe₂ and formyl (Te···O 2.67 Å) also exists in telluronium *Z*-**6**, with DTE adopting an antiparallel configuration (Fig. 2B). Two additional intermolecular chalcogen bonds with triflate anions further stabilize the electron-deficient structure (Te···O 3.00 and 3.08 Å), creating a [2 + 2] complex[48,49].

DFT calculations at M06-2X-D3/def2-TZVP level showed the thermodynamic stabilization of *Z* isomer in relative to *E* isomer for **3**, with the opposite for **1** and **2** (Supplementary Figs. 56–58). To avoid the interference from the environment gas phase computation was run. The rotamer with a short Ch···H (CHO) contact is preferred for *Z*-**1** and *Z*-**2**. Differently, the structure exhibiting Te···O chalcogen bond is the most stable rotamer for *Z*-**3**, *Z*-**4**, and *Z*-**5**, overtaking *E* isomers (Supplementary Figs. 58–60). Chalcogen bond-containing *Z*-isomer of **6** and **7** is significantly energetically favorable over respective *E* isomers (Supplementary Figs. 61 and 62). These results are consistent with crystal structures and demonstrate the capability of chalcogen bonds for controlling over structural preference in the current scaffolds.

### Photoswitching studies of neutral compounds
With the compounds in hands, their photochromic behaviors were next examined in detail (Supplementary Figs. 65–91). Taking the DMSO-*d₆* solution of *E*-**3** at 25 °C as an example, the *Z* isomer (*Z*-**3**) was dominant (96%) at the photostationary state (PSS) when a sample of *E*-**3** was irradiated at 365 nm for 24 min (Fig. 3A and Supplementary Tables 5-6). Interestingly, the photocyclization product *C*-**3** was not detected. Moreover, the ensuing illumination at 450 nm for 40 min restored *E*-**3** (45%) (Fig. 3A). Such bidirectional photoisomerization was repeated multiple cycles, exihibiting reversibility and fatigue-

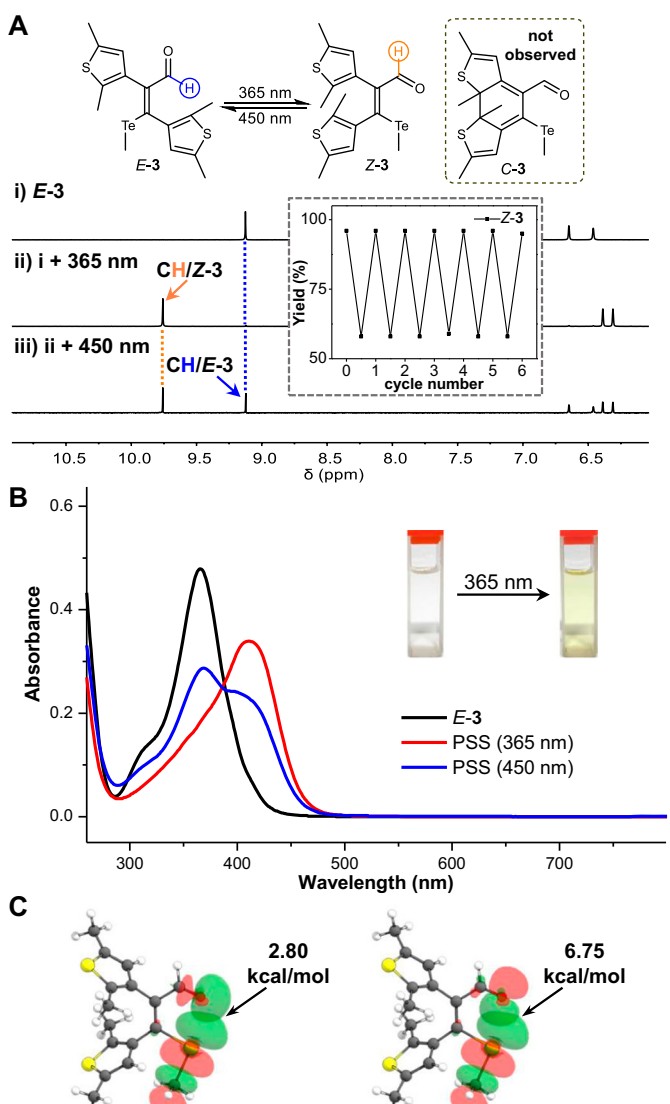

**Fig. 3 | Photoswitching behaviors of tellurium compounds. A** [1]H NMR spectra of *E*-**3** in DMSO-$d_6$ (10 mM, 25 °C, **A**i) upon irradiation with 365 nm (**A**ii) and 450 nm (**A**iii) light, with multiple cycles of bidirectional switching in the inset. **B** Absorbance spectra of irradiation of *E*-**3** in DMSO (60 µM) with 365 nm light and then 450 nm light. **C** Calculated NBO orbitals and stabilization energies (kcal/mol) of n→σ* interactions for chalcogen bond in *Z*-**3**.

revealed an n→σ* stabilization energy of 3.74, 5.69, and 9.55 kcal/mol for chalcogen bonds in *Z*-**1**, *Z*-**2**, and *Z*-**3** (Fig. 3C and Supplementary Fig. 64 and Supplementary Table 3), in consistence with the strongest interaction for Te. The difference in calculated bond length and Wiberg bond index of the ethene bridge (C3-C4) between *Z*/*E* isomers falls in line with enhanced resonance-assisted chalcogen bonding for *Z*-**3** (Table 1 and Supplementary Table 4) as compared to *Z*-**1** and *Z*-**2**. The significant bathochromic shift for *Z*-**3** also echoes enhanced resonance effect (Supplementary Fig. 90).

To gauge the stability of isomers and tune the efficiency of *Z* → *E* isomerization for telluroether compounds, different types of stimuli were employed (Supplementary Figs. 92–103). Unlike conventional *E*/*Z* photoswitches with more thermodynamically favorable *E* isomer, *Z* isomer is more stable for **3**–**5** at 25 °C. *Z*-**3**, *Z*-**4**, and *Z*-**5** remained intact at 25 °C in DMSO-$d_6$ or CDCl₃ (Supplementary Figs. 92–94). Nevertheless, thermal *Z* → *E* isomerization occurred upon heating at 125 °C in DMSO-$d_6$ (Supplementary Fig. 97), with *E*-**3** (60%) overtaking *Z*-**3** upon reaching equilibrium ($t_{1/2}$ around 49 h, Supplementary Table 7). Similar results were obtained with *Z*-**4** at 125 °C in DMSO-$d_6$ (Supplementary Fig. 98). Interestingly, when *Z*-**3** was illuminated with 450 nm light at 120 °C, the *Z* → *E* efficiency was also improved (Supplementary Table 5). Furthermore, the use of acid was able to accelerate configurational isomerization. The addition of methanesulfonic acid (MA) into *Z*-**3** (80 °C) led to the formation of *E*-**3** (58%, Supplementary Fig. 100). When the solvent was changed to CDCl₃, acid-triggered *Z* → *E* conversion even took place quickly (5 min) at room temperature (Supplementary Fig. 101). The identical distribution of *E*/*Z* isomers was obtained after reaching equilibrium when *E*-**3** was subjected to similar treatment (Supplementary Figs. 100 and 101). This is reasonable because of acid-induced activation of the aldehyde and partial loss of double bond character for the olefin, enhancing the rate of *E*/*Z* isomerization in low polar solvent[51]. In a competitive solvent like DMSO-$d_6$ heating is needed along with the acid. Therefore, the use of different stimuli allowed the realization of bidirectional *E*/*Z* switching.

### Photoswitching studies of cationic telluronium

Having achieved the regulation of photoswitching with chalcogen bonding of tellurium, attention was turned to positively charged telluronium (Supplementary Figs. 104-117). The photochemical ring-closure to afford *C*-**6** (42%) occurred along with *Z* → *E* isomerization after *Z*-**6** in DMSO-$d_6$ was irradiated at 425 nm for 10 min (Fig. 4Aii and Supplementary Table 8). The following illumination at 535 nm for 20 min allowed selective addressing (i.e., ring-opening) of *C* isomer to give *Z* isomer (Fig. 4Aiii). As *E*-**6** remained constant (31%) after initial photoirradiation *Z*/*C* photoisomerization was attained (Supplementary Figs. 104 and 105). Similar findings on turning on electrocyclization of *Z*-**6** were obtained in CDCl₃ (Supplementary Fig. 106). UV-vis spectra further verified the photocyclization product (*C*-**6**), giving a broad band around 515 nm (Fig. 4B).

When 2-methyl on the thiophene was changed to phenyl (*Z*-**7**), a mixture of *Z*-**7** and *E*-**7** (86:14) was collected. Gratifyingly, *Z*/*C* photocyclization/cycloreversion accounted for the major pathway, affording *C*-**7** (81%) and *Z*-**7** (93%) upon irradiation under 425 nm (Fig. 4Cii and Supplementary Table 8) and then 650 nm light (c of Fig. 4Ciii). The extended conjugation arising from attached phenyl groups could contribute to the stability of *C*-**7**, as evidenced by the blue color and absorbance spectra reaching around 700 nm (Fig. 4D). The isolation and characterization of *C*-**7** further confirmed 6π-electrocyclization (Supplementary Figs. 53 and 54). Multiple cycles of bidirectional *Z*/*C* photoswitching with fatigue-resistance was successful (Fig. 4C and Supplementary Fig. 112). With the percentage of *E*-**7** minor and unchanged (7%) during illumination cycles the efficiency of photochemical ring-closure/ring-opening was significantly enhanced. It is notable that the current

resistance over 12 consecutive irradiations with light of 365 nm and 450 nm (Supplementary Fig. 66). *E*/*Z* photoswitching was feasible for *E*-**3** in CDCl₃ (Supplementary Table 5 and Supplementary Fig. 67). Photochromic features of **3** were further confirmed by absorbance spectra, interconverting between bands of 365 (*E*-**3**) and 412 nm (*Z*-**3**) (Fig. 3B and Supplementary Figs. 69–71). The realization of *E*/*Z* configurational switching was also found with **4** and **5** (Supplementary Table 5 and Supplementary Figs. 72–82). In sharp contrast, *Z*-isomer and ring-closure product (*C*) of S-containing *E*-**1** or Se-containing *E*-**2** were observed after irradiation at 313 nm, indicating the involvement of both *E*/*Z* isomerization and photocyclization/cycloreversion (*Z*/*C*) mechanisms (Supplementary Figs. 83–88).

We attribute the striking differences between Te derivative (**3**–**5**) and S and Se derivative (**1**, **2**) with resonance-assisted chalcogen bonding: in addition to the stabilizing role of the stronger chalcogen bond of Te on *Z* configuration the enhanced resonance effect renders C=C bond partial single bond character, thus suppressing the electrocyclization pathway (Table 1). Natural bond orbital (NBO) analysis[50]

**Table 1 | Resonance-assisted chalcogen bonding with calculated bond length (Å) and Wiberg bond index (WBI) as well as measured absorption maximum ($\lambda_{max}$, nm) for Z/E isomers of 1–7**

*resonance-assisted chalcogen bond*

| Compound | Ch | X | C3-C4 (E) | | E $\lambda_{max}$ (nm)[a] | C3-C4 (Z) | | Z $\lambda_{max}$ (nm)[a] |
|---|---|---|---|---|---|---|---|---|
| | | | Len (Å) | WBI | | Len (Å) | WBI | |
| 1 | SMe | Me | 1.353 | 1.679 | 313 | 1.362 | 1.624 | 323 |
| 2 | SeMe | Me | 1.349 | 1.707 | 329 | 1.359 | 1.638 | 340 |
| 3 | TeMe | Me | 1.346 | 1.735 | 365 | 1.359 | 1.641 | 412 |
| 4 | TePh | Me | 1.345 | 1.743 | 358 | 1.358 | 1.646 | 401 |
| 5 | TeMe | Ph | 1.348 | 1.735 | 368 | 1.359 | 1.641 | 410 |
| 6 | TeMe$_2$ | Me | 1.339 | 1.808 | / | 1.348 | 1.729 | 296 |
| 7 | TeMe$_2$ | Ph | 1.338 | 1.809 | / | 1.348 | 1.729 | 300 |

[a]In DMSO.

telluronium scaffold represents one all-visible-light photoswiching system[52–54].

The n→σ* interacting energies were found to be 11.35 and 11.19 kcal/mol for chalcogen bonds of Z-6 and Z-7 (Fig. 4E). However, the resonance effect is weakened, as revealed by a shorter bond length of 1.348 Å and a larger bond order of 1.729 for the ethene bridge (C3-C4) in Z-6 than those for Z-3 (Table 1). Analogous trends were observed for Z-7 and Z-5. Despite strengthening chalcogen bond from tellurium to telluronium for the stabilization of Z isomer, resonance donation from telluronium to aldehyde would be sabotaged, thus maintaining double bond character and the capability of photocyclization. DFT calculations revealed the presence of intramolecular chalcogen bonding in C-6 and C-7 (Fig. 4E and Supplementary Figs. 61 and 62). The varying locations of HOMO and LUMO orbitals for Z-3 and Z-6 match their photoswitching patterns (Supplementary Fig. 63). Tying it all together, the domination of different photoisomerization pathways (E/Z and Z/C) can be accomplished by manipulating the strength and underlying mechanism of chalcogen bonds.

### Dynamic covalent chemistry gated photoswitching

Encouraged by unique photochromic behaviors of tellurium and telluronium-derived DTEs, their reversible interconversion was pursued for gating photoswitching with dynamic covalent reactions (DCRs)[55–60] (Fig. 5). Toward this end, dynamic nucleophilic substitution reactions were attempted[61–64]. A mixture of Z-3 and methyl triflate (1.8 equiv.) in CDCl$_3$ gave Z-6 (-100%), with the appearance a new peak around 9.75 ppm (Fig. 5A and Supplementary Figs. 118–119). The yield of Z-6 was dependent on the amount of methyl triflate, suggesting an equilibrium reaction for the formation of telluronium (Supplementary Fig. 120). The use of tetrabutylammonium iodide (TBAI) allowed the realization of the reverse reaction of Z-6 to restore neutral tellurium switch Z-3 (Fig. 5A and Supplementary Fig. 121). Further addition of methyl triflate again afforded Z-6. The dynamic covalent interconversion between Z-3 and Z-6 was thus achieved in a closed-loop fashion, as also the case for Z-5 and Z-7 (Supplementary Figs. 122–124).

To shed light on the mechanism of transformation of telluronium, a series of tetrabutylammonium salts of different anions were screened (Supplementary Figs. 125–132). Upon the addition of TBAI into Z-6 there was an upfield movement of the aldehyde peak to 9.82 ppm, with an opposite shift for thiophene CH near telluronium (Fig. 5B and Supplementary Fig. 125). Moreover, the intermediate gradually faded to form Z-3 in quantitative yield within 65 min. These observations are explained as following: anion binding to electron-poor telluronium cation occurs first to afford adduct Z-6-I, which then converts to Z-3 via a reductive-elimination pathway[65,66]. The interpretation was further backed with chloride and bromide salts, which decelerated the kinetics of substitution reactions in relative to iodide (Supplementary Figs. 127–131).

DCRs were then coupled with photoswitching as a means to select photoisomerization pathways and accordingly modulate component distribution. For example, the reaction of Z-7/E-7 (86:14) with TBAI to form Z-5/E-5 followed by the irradiation at 365 nm afforded nearly exclusively Z-5 via E/Z switching (Fig. 5C and Supplementary Fig. 135). The subsequent DCR with methyl triflate led to Z-7 and turned on the photocyclization. The illumination under 425 nm light allowed mainly the creation of C-7, with pericyclic ring-opening at 650 nm to complete the reaction cycle. The in situ manipulation of switching patterns was also achieved with 3 and 6 (Supplementary Figs. 136 and 137). Therefore, a combination of distinct photoswitching behaviors and dynamic covalent reactivity enabled the control over photoisomers on purpose in a multistep manner.

### Photoaddressed rewritable materials

Finally, to showcase the utility light-mediated rewritable materials were constructed. The controllable patterning and writing would be of interest to data storage and anti-counterfeiting coding[67,68]. A yellow filter paper was prepared by employing a solution of Z-3 in chloroform (Fig. 6a). The illumination of the filter paper covered with customized geometric pattern mask under 425 nm light led to neglectable changes. Nevertheless, the placement of Z-3 into methyl triflate vapor toggled on Z/C photoswitching due to the creation of Z-6. As a demonstration, through choosing photomasks various red patterns (i.e., C-6) were printed on the same filter paper via visible light irradiation (Fig. 6a and Supplementary Fig. 138). Such "write-and-erase" cycles were repeated multiple times by alternating between 425 and 535 nm light illumination, indicating excellent reversibility and high spatial resolution.

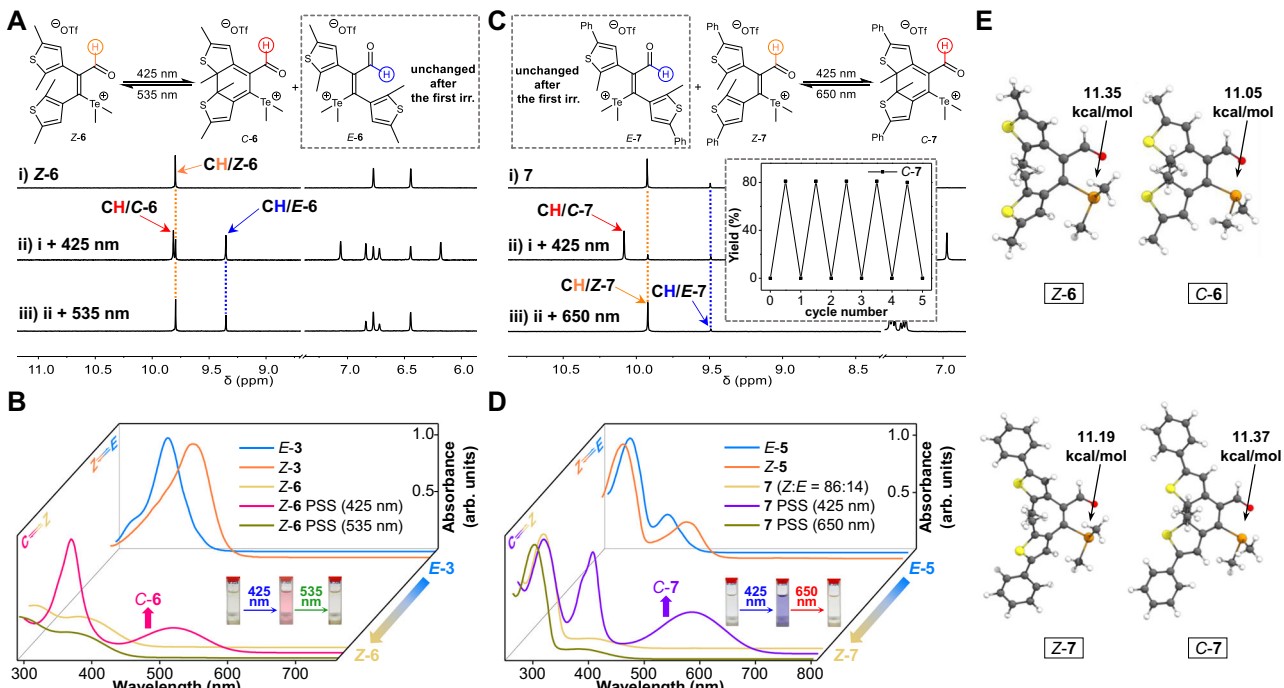

**Fig. 4 | Photoswitching behaviors of telluronium compounds. A** ¹H NMR spectra of Z-**6** in DMSO-$d_6$ (10 mM, 25 °C, **Ai**) upon irradiation (irr.) with 425 nm (**Aii**) and 535 nm (**Aiii**) light. **B** Absorbance spectra of Z-**6** in DMSO (100 μM) upon irradiation with 425 nm and then 535 nm light, with the spectra of E/Z photoswitching of E-**3** in DMSO (60 μM) at 365 nm included for comparison. **C** ¹H NMR spectra of **7** in DMSO-$d_6$ (10 mM, 25 °C, **Ci**) upon irradiation (irr.) with 425 nm (**Cii**) and 650 nm

(**Ciii**) light, with multiple cycles of bidirectional switching shown in the inset. **D** Absorbance spectra of **7** in DMSO (50 μM) upon irradiation with 425 nm and then 650 nm light, with the spectra of E/Z photoswitching of E-**5** in DMSO (60 μM) at 365 nm included for comparison. **E** Calculated structures of Z-**6**, Z-**7**, C-**6**, and C-**7**, with NBO stabilization energies (kcal/mol) of n→σ* interactions for chalcogen bonds listed.

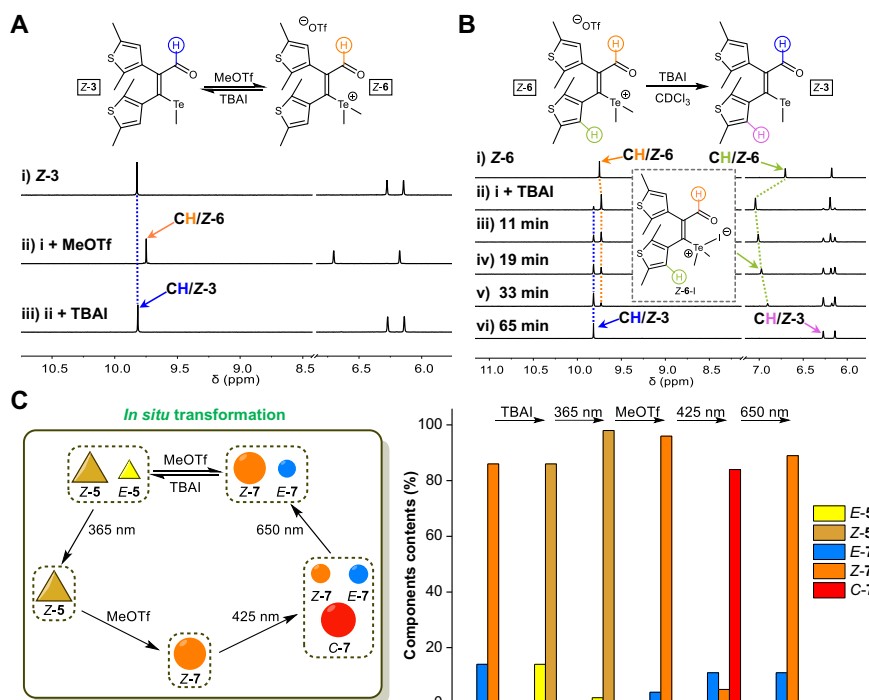

**Fig. 5 | Regulation of photoswitching with reversible nucleophilic substitution reactions. A** ¹H NMR spectra of Z-**3** in CDCl₃ (10 mM, 25 °C, **Ai**) upon reactions with methyl triflate (MeOTf, 1.8 equiv., **Aii**) and then tetrabutylammonium iodide (TBAI, 2.1 equiv., **Aiii**). **B** ¹H NMR spectra of the reaction of Z-**6** (10 mM, 25 °C) and

tetrabutylammonium iodide (TBAI, 1.3 equiv.) in CDCl₃ to create Z-**3** at varied time, with the intermediate Z-**6**-I (Z-**6** with iodide ion) shown. **C** The control of the distribution of photochromic states via dynamic covalent reactions and light irradiation within one multistep cycle (5% experimental error).

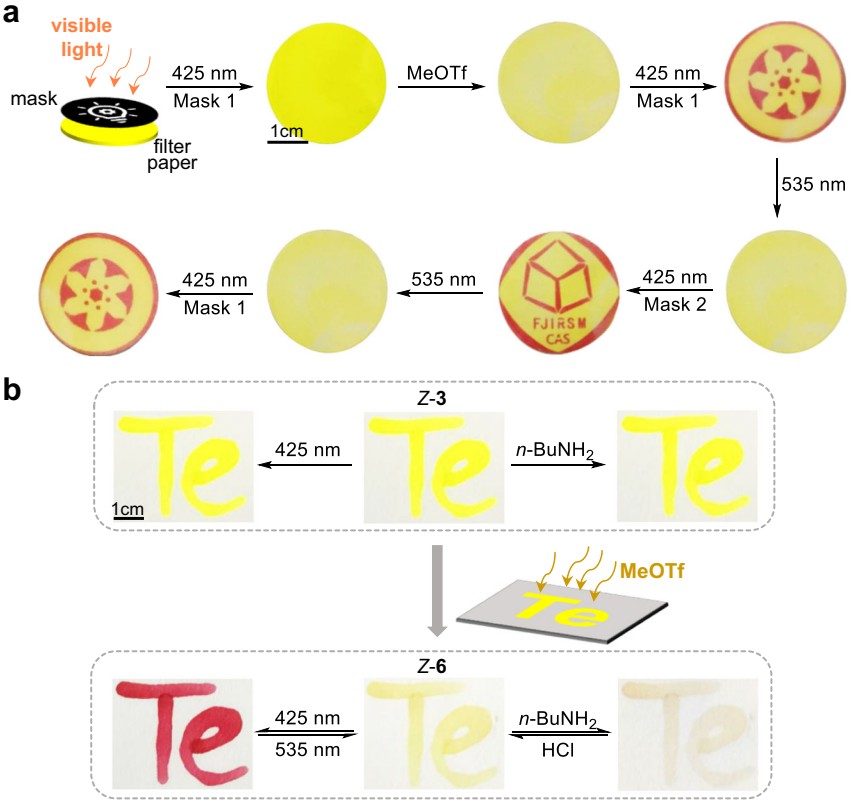

**Fig. 6 | Light-mediated rewritable materials. a** The manipulation of *Z*-**3** loaded filter paper with dynamic nucleophilic substitution and photoswitching for varied patterns. The logos of Chinese Academy of Sciences (CAS) and Fujian Institute of Research on the Structure of Matter (FJIRSM) were used as pattern masks with permission. **b** The manipulation of writing of *Z*-**3** as letters "Te" on a filter paper with photoswitching and dynamic imine chemistry, triggered by reversible nucleophilic substitution.

To further pursue multi-stimuli responsiveness and smart patterning dynamic imine chemistry was studied by taking advantage of unique chalcogen bonding patterns (Supplementary Figs. 139–152). The kinetics of imine formation was tracked in CDCl₃. The reaction of *Z*-**3** and 1-butylamine to give imine *Z*-**3A** proceeded very slowly (33% *Z*-**3A** after 65 h) (Supplementary Fig. 139). However, the rate for the corresponding telluronium *Z*-**6** was significantly enhanced, taking only 12 min for the quantitative formation of imine *Z*-**6A** (Supplementary Fig. 143). While resonance-assisted chalcogen bonding deactivates the aldehyde in *Z*-**3** with partial single bond character, neighboring chalcogen bond in *Z*-**6** would activate the formyl group in the form of intramolecular Lewis acid catalysis without the implication of resonance effect from telluronium[69]. For the poof-of-concept letters "Te" were written on a filter paper with *Z*-**3** (Fig. 6b). The illumination at 425 nm as well as the exposure to 1-butylamine vapor afforded virtually no color change, and again the treatment of *Z*-**3** with methyl triflate vapor to form *Z*-**6** triggered switchable writing. After irradiation at 425 nm *C*-**6** and thus the red color appeared, with the color further erased upon exposure to 535 nm light. Moreover, the vapor of 1-butylamine induced fading of yellow color of *Z*-**6** arising from imine formation (Fig. 6b and Supplementary Fig. 149). The use of HCl vapor allowed the regeneration of the aldehyde and thus the recovery of the color. The changes of *Z*-**6** in response to different stimuli were further confirmed by NMR studies (Supplementary Fig. 150). By employing only a single filter paper light and pH stimuli induced three-state switching system (light tan-yellow-red) was obtained, with methyl triflate serving as an activating reagent for another handle of control. Similar results were obtained with *Z*-**5** (Supplementary Figs. 151 and 152). Through leveraging photochromic features and dynamic covalent reactivity switchable rewritable materials were readily fabricated, showing the potential in information encryption and multilevel logic gates.

In this work, we report a versatile platform of regulating multistep photoswitching through chalcogen bond-containing open-bridged dithienylethenes (DTEs). A series of chalcogen derived light switches were constructed in a modular way, and the role of chalcogen bonding in selecting the pathways of *E/Z* isomerization and photocyclization/cycloreversion was investigated. Efficient bidirectional *E/Z* photoswitching was accomplished with neutral tellurium, different from cationic telluronium for which photocyclization/cycloreversion was attained. The reversible nucleophilic substitution further enabled the modulation of chalcogen bonds and the gating of photochromism on purpose. To show the utility photomediated multistate rewritable materials were created. This work demonstrates the capability of noncovalent bonding bridged photoswitches and enriches the toolbox of multistep photoswitches. The strategies and results described should find applications in dynamic interactions, molecular switches, anti-counterfeiting coding, and responsive materials.

## Methods

### General
<sup>1</sup>H NMR and <sup>13</sup>C NMR spectra were recorded on a 400 MHz Bruker Biospin Avance III spectrometer or a 400 MHz JEOL JNM-ECZ400S spectrometer. The chemical shifts (δ) for ¹H NMR spectra, given in ppm, are referenced to the residual proton signal of the deuterated solvent. Crystallographic data was collected on a Mercury single crystal diffractometer at room temperature. The structures were solved with direct methods by using OlexSys or SHELXS-97 and refined with the full-matrix least-squares technique based on F2. The UV-vis spectra were measured on a PerkinElmer Lambda 365 spectrometer.

All other reagents were obtained from commercial sources and were used without further purification, unless indicated otherwise.

## Irradiation experiments

The irradiation experiments were performed by using a 20 W LED lamp (640–650 nm) equipped with a narrow pass filter of 650 nm, a 15 W LED lamp (590–600 nm) equipped with a narrow pass filter of 600 nm, a 15 W LED lamp (530–540 nm) equipped with a narrow pass filter of 535 nm, a 20 W LED lamp (440–450 nm) equipped with a narrow pass filter of 450 nm, a 20 W LED lamp (420–430 nm) equipped with a narrow pass filter of 425 nm, a 10 W LED lamp (360–370 nm) equipped with a narrow pass filter of 365 nm, and a CEL-HXF300 xenon lamp with a bandpass filter at $313 \pm 10$ nm, respectively. The illumination of solution samples was conducted until the photostationary state (PSS) was reached, as tracked by using NMR or UV-vis spectroscopy. The ratio of the E/Z/C forms was determined through $^1$H NMR analysis.

## Dynamic covalent reactions

Dynamic covalent reactions were performed in situ in $CDCl_3$ without isolation and purification. For telluronium formation, to a stirred solution of an aldehyde (10 mM, 1.0 equiv.) in $CDCl_3$ (0.60 mL), was added MeOTf (18 mM, 1.8 equiv. or 30 mM, 3.0 equiv.). For telluronium decomposition, to a stirred solution of an aldehyde (10 mM, 1.0 equiv.) in $CDCl_3$ (0.60 mL), was added tetrabutylammonium salt of different counteranion (13 mM, 1.3 equiv.). For imine formation, to a stirred solution of an aldehyde (10 mM, 1.0 equiv.) in $CDCl_3$ (0.60 mL), was added 1-butylamine (14 mM, 1.4 equiv.). The mixture was tracked by $^1$H NMR until the equilibrium was reached.

## DFT calculations

Geometry optimization and frequencies calculations were performed by using Gaussian 09 (G09) packages[70]. The DFT method and basis set of M06-2X-D3/def2-TZVP with an ultrafine integration grid were employed during the optimization and frequencies analysis. The default settings of def2-TZVP for Te in G09 was made up of relativistic effective core potential and the Stuttgart-Dresden effective core potential frozen core basis set MWB28. All the geometries were determined without imaginary frequencies by frequency analysis. The conformer search was done for compounds **1**–**3**, and **6**. Five major minima were thus found: Z forms with/without chalcogen bonding, E form, and ring forms C with/without chalcogen bonding (Supplementary Figs. 56–58 and 61). For the simplicity, only the most stable E/Z/C conformers were selected for the calculation of other Te containing compounds (**4**, **5**, and **7**, Supplementary Figs. 59, 60 and 62). The NBO analysis[50] was implemented by NBO 3.1 module in G09. The electronic density cubes for electrostatic potential map (ESP) and Atoms in Molecules (AIM) were generated by Multiwfn 3.80 and presented by VMD 1.90.

## Photoswitchable filter paper preparation and patterning

Whatman filter paper was soaked in the solution of Z-**3** (5.0 mM in $CHCl_3$) for 30 seconds, followed by evaporation to dry. The filter paper was exposed to different wavelengths of light with different customized geometric pattern mask and vapor generated from MeOTf. Photoswitchable filter paper was prepared by writing a stock solution of Z-**3** or Z-**5** (10.0 mM in $CHCl_3$) onto Whatman filter paper with a fine brush, followed by evaporation to dry. The filter paper was exposed to different wavelengths of light and different vapors generated from MeOTf, aqueous n-$BuNH_2$ (40% w/w), and aqueous HCl (30% w/w).

## Data availability

The data supporting the findings of this study are available within the paper and its Supplementary Information. Crystallographic data for the structures reported in this article have been deposited at the Cambridge Crystallographic Data Centre, under deposition no. CCDC 2249588 (E-**4**), 2249589 (Z-**4**), 2249590 (Z-**5**), 2249591 (Z-**6**), 2260917 (E-**2**), and 2260918 (E-**3**). Copies of the data can be obtained free of charge via http://www.ccdc.cam.ac.uk/data_request/cif. All other data are available from the authors upon request.

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

## Acknowledgements

We thank National Natural Science Foundation of China (92156010, L.Y.; 22071247, L.Y.; 22101283, H.Z.; 22101284, H.Y.), the Strategic Priority Research Program (XDB20000000, L.Y.) and the Key Research Program of Frontier Sciences (QYZDB-SSW-SLH030, L.Y.) of Chinese Academy of Sciences, Natural Science Foundation of Fujian Province (2020J06035, L.Y.; 2022J05085, H.Y.), and Fujian Science & Technology Innovation Laboratory for Optoelectronic Information of China (2021ZR112, L.Y.) for support.

## Author contributions

L.Y. conceived of the idea and directed the research. S.J. designed the protocols, performed the experiments, and analyzed the data. H.Y. conducted density functional theory calculations. P.H. and X.L. participated in organic synthesis. L.Y. and S.J. wrote the manuscript. All authors discussed the results and revised the manuscript.

## Competing interests

The authors declare no competing interests.
