## [Peer Review File · Nature Communications]

Selection of Isomerization Pathways of Multistep Photoswitches by Chalcogen BondingREVIEWER COMMENTS

Reviewer #1 (Remarks to the Author):

The development of new photochromic switches is of notable importance, and control over multistep and multistage photoswitching is challenging. The authors reported a series of chalcogen bonding constrained open dithienylethenes (DTEs) and studied photoswitching features in detail. The bidirectional E/Z photoswitching was found with neutral tellurium, and cationic telluronium allowed the realization of photocyclization/cycloreversion. The critical role of chalcogen bonding in selection from multistep sequence of photoisomerization was elucidated. Reversible nucleophilic substitution reactions were further used to gate the photoswitching and component distribution via interconversion between tellurium and telluronium. The potential in multistate rewritable materials with high spatiotemporal resolution was also demonstrated. The experimental work was conducted with vigor, and the strategy of noncovalent bond bridged photoswitches should set the stage for future studies. I recommend the acceptance once the following minor issues are addressed:

1. Besides the difference induced by the type of chalcogens (S, Se, and Te), does the substituent on Te affect photoswitching?
2. The higher efficiency of photochemical ring-closure for Z-7 than Z-6 should be explained. How about thermal stability of C-7 and C-6?
3. Considering dynamic nucleophilic substitution reactions, is it possible to trap the anion binding intermediates, such as Z-6-I?
4. Selected light wavelengths were utilized for photoirradiation. How about the effect of changing wavelengths on photoswitching behaviors?
5. Z-3 and Z-6 appeared to afford sharply different reactivity toward amine vapors. Please check their reactions in solution for the validation.

Reviewer #2 (Remarks to the Author):

In this work the team of You describes the synthesis, experimental and theoretical analysis and application of a novel diarylethene-based photoswitch with four state behavior within a single switching unit. The diarylethene photoswitch is a special derivative as it is not ring-fused at the central double bond. This unlocks the option for additional cis-trans isomerization next to ring opening and closing upon light irradiation. However, it also causes problems because of the inherent difficulty to selectively address only one photo process at the time. The authors address this challenge by using deliberate and tunable chalcogen bonding via introduction of a tellurium-ether site and an aldehyde at the central double bond. In the neutral state of the tellurium-ether selective cis-trans photoisomerization takes place and the authors are able to obtain >90% switching to the Z isomer and up to 67% of the E isomer. This selectivity changes when the tellurium-ether is methylated to the corresponding telluronium cation, which enhances the chalcogen bonding and abolishes cis-trans photoisomerization. Now electrocyclic

ring closure and opening is observed with the corresponding strong color changes in very high overall yields. Methylation can be reversed by adding TBAI, which also renders the control over photochemical pathways reversible. The authors demonstrate this in a cyclic experiment cycling between essentially three different states of the system. They further provide an application for reversible writing and erasing of information on paper using photomasking. When reversible dynamic imine-chemistry is introduced it is possible to bleach and recover the yellow color of the telluronium open Z isomer.

Overall this study provides a very intriguing and well working concept for the establishment and control of multi-state photoswitching without resorting to oligomerization of switching units. The findings of this study represent a very high degree of conceptual novelty and performance and explore new and exciting possibilities for multi-addressability within small molecular building blocks. As stated above, gaining precision control over alternative types of photochemistry within the same molecular building block is highly challenging and I congratulate the authors to their very nice results. For these reasons I am happy to recommend publication of this work in Nature Communications, however a number of important points need to be addressed before, which are outlined below.

Major points:

- One major point is the still lacking Z to E photoisomerization in almost all derivatives. Only for derivative 4 can more E isomer be accumulated than Z. This is of course a pity, as all the other three photo processes are much more effective. It is also curious as the photochromism should allow for very efficient Z to E photoisomerization with wavelengths >420 nm (e.g. Figure S48). I would strongly urge the authors to try 1. Longer wavelengths of light for this photoisomerization, different solvents with different properties, e.g. protic ones, very apolar ones like alkanes etc. It is also curious if not temperature effects play a major role here, and if not photoisomerizations at lower or higher temperatures are found more effective in this regard
- Another major point is the mentioned imine-chemistry. This is interesting as a color effect, but no details are given about the degree of imine-formation, the thermal stability of isomers or the photochemistry of the imine-products. Please provide more information in this regard and also explain more in detail in the text how this reversible functionalization affects the overall performance.
- What are the quantum yields for the different photo processes?
- The assignment of isomers is not always straight forward. I miss a thorough NMR analysis (NOEs, HSQC, HMBC etc.) for isomer identification.
- Please add a table with all measured and calculated important data summarized. That should also include thermal parameters, which are not yet extracted from the heating experiments. Thermal data that need to be reported are the different relative isomer stabilities (ΔG s from the equilibrium isomer distribution reached after prolonged heating in the dark) and the associated first order kinetics (either with or without entering equilibria) and resulting Gibbs energies of activation ΔG^* and half-lives. Where this is not possible lower limits for these values should be given.
- extinction coefficients are also missing for the photoswitches and their isomers.

Minor points:

- The manuscript needs some English corrections, many sentences are hard to read and are not easy to understand.
- Figure captions are hard to understand because letters were used in capital and small letters. Better use A for the image sections and then i, ii and so on for within the image sections. The small letters in the image sections are arranged somewhere in the text and it is not clear how much content belongs before or after them. Please always put them at the beginning, e.g. Ai) xxx Aii) etc.
- Line 10: write: „and are challenging to control.”
- All NMR spectra that are stacked, whether in SI or Paper Draft, should be reported in invert order. Because the first measured NMR is always listed as the final NMR and one must read from bottom to top.
- The mixed nomenclature of E/Z and cis/trans needs to be unified, please decide for one of the two
- Naming of isomers and mechanisms is not well selected in general. Cycloreversion is abbreviated with C just as the closed form is abbreviated with C. Z also stands once for the Z isomer and once for photocyclization. It is best to use the term E/Z isomerization and label the isomers Z/E and photocyclization/cycloreversion is PC/CR and label the isomers o (open) and c (closed).
- Line 10: write: “Here we demonstrate a multistep sequence of trans/cis...”
- Line 13: write: “..on open ethene bridged dithienylethenes offer a...”
- Line 14: write: “While bidirectional E/Z photoswitching is dominated by neutral...”
- Line 16: Write “..the creation of cationic telluronium enables the...”
- Line 17: allows
- Line 37: Write: “Information encoding”
- Line 38: „type of blossoming“ not scientific
- Line 39-40: “...”, which can be utilized in...”
- Line 45: „Adjacent chalcogen bonds“ chalcogen bond
- Line 54, 55: Please define R1 and R2 in Fig. 1 a
- Line 57: „chalcogen-containing dithienylethene photoswitches“ This is confusing naming, because the sulfur within the thiophenes of the parent dithienylethenes is also a chalcogen. This should be more clearly described.
- Line 71: X: The X stands for chalcogen, which was previously abbreviated as Ch. This should be defined uniformly
- Line 72: I would rather write a “Vilsmeier-Haack-Arnold formylation type” reaction, because it is not that classical.
- Fig 1: Please add all the stimuli at all arrows.
- Fig 2: Please give more details about synthesis including reagents into the Figure.
- Fig 2: The reversibility at 2A c that you proclaimed in Figure 1 should be highlighted in this Figure too.
- Line 81: TeMe2: TePh because it is Z-4
- Line 89 – 92: Reaction condition should be with A in the reaction overview not with B
- Page 6 line 102 – 114: It is a bit confusing, as there is a permanent change between NMR and UV-vis data.
- Line 103-106: I am missing the temperatures you measured the NMRs when reaching the PSS.
- Line 107: Why was the cycle experiment for compound 4 in DMSO not conducted as it is the best candidate for photoswitching?
- Line 108: “Please write “exhibiting reversibility and fatigue resistance over 12 consecutive irradiations

with light of 365 nm and 450 nm.”

- Please enter the NMR spectra in invert order.
- C: Len unit is missing, λ_{\max} is smaller than the rest
- Line 108: 45% is not a great reversibility
- Line 115: Te derivative (3-5) and S and Se derivative (1, 2)
- Line 128: write how many cycles instead of multiple
- Line 132-141: This section is not fully stringent in the argumentation. I think what happens after addition of acid is that the double character at the open bridge area is somewhat lost, and this is why the sterically more favoured E isomer is formed. Of course this isomerization takes more time without adding acid. But it is a bit counter intuitive for me, that more polar solvents lead to greater thermal stability while addition of acid lowers it? How do you explain that?
- Page 7 line 133 – 136: Compounds Z-3, Z-4 and Z-6 were heated to 80 °C in DMSO for 12 h only and there is no thermal isomerization happening. One should then test a higher temperature. By adding a very strong acid at 80 °C, 58% E is formed according to the text, but according to the NMR label only after 90 min at 80 °C? However, on page 8 line 141 it is then claimed that switching can be done with heat, light and acid. In fact, no evidence for thermal isomerization is given. -> Here, conclusions are drawn without providing more detailed evidence and, most importantly, acid and heat were used simultaneously and it is not clear which change is influenced by the heat and which by the acid
- Line 153: 425 nm is missing
- Line 157: fatigue resistance is shown, but for this rather low cycle number I would now say it is high.
- Line 200 – 201: What is Z-6-I? Please define and then also name the molecule in Fig 5
- Fig. 5: please add the NMR data for the experiments shown in C for derivatives 5 and 7, these data are crucial to evidence the claims
- Fig. 6 and imine chemistry – as stated above this part needs rework. It is not clear what imine chemistry is referred to and what is meant by “lock/unlock” in the Fig. 6. As it is, imine formation simply causes a color change without isomerization in the case of the telluronium form but not in the tellurium form. “Locking/unlocking” are therefore misplaced terms here that should be removed.
- Line 262: anti-counterfeiting
- Line 273: 530 – 540 nm
- Line 295: (C): C

SI:

- From your table S5 it is not clear to me what you experimentally did. You started with the E isomers and enriched the Z-isomers, and then you irradiated the solution of the Z-enriched species to enrich the E-isomers? Please clarify the order of irradiation events and starting and end points.
- In some cases, for example for compound 3, the E and Z isomer assignments show some mistakes. It is written E-3 but for, I guess, the Z-isomer is just written 3.
- It is not clear, how the E and the Z isomers were assigned, as ¹H NMR NOE experiments are missing. I guess NOE experiments were used to determine whenever it is the Z or the E isomer, but as it is crucial for this work it should be described in the SI
- Do you mean Z-4 for 4 at the NMR section on page S6?
- Please add to the Figure captions of the NMR spectra shown in the SI what the measuring temperature

was, and that you measured on 400 MHz and 141 MHz.

- Consistency: Isomeric ratio given sometimes 86/14 and sometimes 7:93 in the SI
- Fig. S1 – S 30 Scale labeling with f_1 (ppm) is not correct, measurement temperature and resonance frequency is not provided, sometimes spectra are not shown up to 0 ppm
- Fig. S44 Is this really an isosbestic point?
- Fig. S46 – S47 What is the signal at 7.75 ppm? A side product?
- Fig. S50 Is this really an isosbestic point?
- In S71 one can see a signal at 7.10 that is partially hidden with the figure on top
- Fig. S64 B and c: What are the new signals? New species?
- Fig. S66 and S68 This is not really a evidence that no thermal isomerization takes place
- Fig. S67 and S69 What are these new signals?
- Table S6 Traces of c, but E and Z together yield 100%?
- Fig. S77 and S78 Sometimes 86/14 and sometimes 7:93.
- Fig. S79 What are the new signals?
- Fig. S93 What are the new signals?

Reviewer #3 (Remarks to the Author):

In the manuscript, the authors describe the synthesis and study of photoswitches that exhibit different switching properties depending on substituents. In the presence of a neutral tellurium substituent, E/Z photoisomerization is found. If the tellurium atom is converted to a cationic species, mainly a photocyclization/cycloreversion by irradiation takes place. As this conversion is reversible, the whole system represents a multistep photoswitch.

The experiments are well thought out and well performed and the field itself is extremely exciting. Nevertheless, unfortunately, I cannot recommend acceptance for publication in Nature Communications due to the following reasons:

1) The first issue is the lack of novelty. It is already known that photoswitches can be affected by chalcogen bonds and thus one can regulate the switching ability (reference 40). In addition, it has already been shown that thiophenyl-substituted alkenes can be switched differently depending on the wavelength used (reference 23). The percentage of switching found there (Figure 5a) partly exceeds the distribution of isomers in the PSS described in this manuscript.

2) The second issue concerns the extent of switching. Switches 3 and 5, which can also be switched as cations and thus represent the actual multistep photoswitches, show only minor switching behavior. In no case was an isomerization of more than 45% possible.

3) The third issue relates to the interpretation of the data. The authors explain the different behavior of

the switches by the different chalcogen bond strengths. Switches 6 and 7 have very similar structures; specifically, the chalcogen bond strength should be almost identical for both. The NBO stabilization energies determined by the authors are 11.4 and 11.2 kcal/mol, respectively. Nevertheless, the two systems show different switching behavior. For example, E/Z isomerization also takes place in compound 6. However, this does not occur with compound 7. If the strength of the chalcogen bond influences the switching behavior, these results are difficult to explain. The difference here probably stems from the different absorption behavior, so that the ratio of the isomers in the PSS is different for 6 and 7 (cf. reference 23).

September 20, 2023

Dear Reviewers,

Thank you very much for reviewing manuscript NCOMMS-23-30805. We have uploaded our revised manuscript entitled “**Selection of Isomerization Pathways of Multistep Photoswitches by Chalcogen Bonding**” that we wish to have considered for publication as an article in *Nature Communications*.

The reviewers provided very helpful comments, and we have made every effort to address the comments and improve this manuscript (the changes/additions are highlighted in red in a copy of the manuscript as Supporting Information). I have so indicated in my point-by-point response and revision summary, which follows.

Responses to comments by Reviewer 1:

Comment: The development of new photochromic switches is of notable importance, and control over multistep and multistage photoswitching is challenging. The authors reported a series of chalcogen bonding constrained open dithienylethenes (DTEs) and studied photoswitching features in detail. The bidirectional *E/Z* photoswitching was found with neutral tellurium, and cationic telluronium allowed the realization of photocyclization/cycloreversion. The critical role of chalcogen bonding in selection from multistep sequence of photoisomerization was elucidated. Reversible nucleophilic substitution reactions were further used to gate the photoswitching and component distribution via interconversion between tellurium and telluronium. The potential in multistate rewritable materials with high spatiotemporal resolution was also demonstrated. The experimental work was conducted with vigor, and the strategy of noncovalent bond bridged photoswitches should set the stage for future studies. I recommend the acceptance once the following minor issues are addressed.

Response: Thanks a lot for the reviewer’s comments and suggestions on our manuscript.

Comment: 1. Besides the difference induced by the type of chalcogens (S, Se, and Te), does the substituent on Te affect photoswitching?

Response: Thanks a lot for the reviewer’s comments. When the methyl on Te (**3**) was changed to phenyl (**4**), the efficiency of *E*→*Z* isomerization was slightly reduced (~90% *Z*, Table S5), and the efficiency of *Z*→*E* isomerization was increased (62% *E* in DMSO-*d*₆, 67% *E* in CDCl₃).

Comment: 2. The higher efficiency of photochemical ring-closure for *Z*-**7** than *Z*-**6** should be explained. How about thermal stability of *C*-**6** and *C*-**7**?

Response: Thank the reviewer for the suggestion. The higher efficiency of photocyclization of *Z*-**7** (X = Ph) than *Z*-**6** (X = Me) is due to the stability of *C*-**7** arising from extended conjugation induced by the phenyl group on thienyl. Through

tracking ^1H NMR spectra, *C-6* and *C-7* (obtained from individual irradiation of *Z-6* and *Z-7*) remained stable in $\text{DMSO-}d_6$ solution at 25 °C for more than three days (Figure S99 and S106).

Comment: 3. Considering dynamic nucleophilic substitution reactions, is it possible to trap the anion binding intermediates, such as *Z-6-I*?

Response: Thank the reviewer for the suggestion. We have tried to isolate *Z-6-I*, but unsuccessful so far. The dynamic nature of chalcogen bonding between *Z-6* and iodide renders *Z-6-I* unstable. The movement of ^1H NMR peaks points to the formation of adduct *Z-6-I* (Fig. 5B).

Comment: 4. Selected light wavelengths were utilized for photoirradiation. How about the effect of changing wavelengths on photoswitching behaviors?

Response: Thanks a lot for raising the issue. We agree that the selection of wavelengths is crucial to the performance of photoswitches, as Reviewer 2 also pointed out. For neutral tellurium compounds, such as **3**, the change in wavelength (450 nm, 465 nm, and 475 nm) does not increase *Z*→*E* photoisomerization efficiency (Figure L4). For the cationic telluronium, photocyclization can also occur with UV light (313 nm), with the isomerization efficiency (77% *C-7*) slightly lower than that of visible light irradiation (81% *C-7*) (Figures L9 and L10)

Comment: 5. *Z-3* and *Z-6* appeared to afford sharply different reactivity toward amine vapors. Please check their reactions in solution for the validation.

Response: Thank the reviewer for the suggestion. We have performed imine formation in solution and added dynamic imine-chemistry in the main text and SI according to the reviewer's advices (Figures S135-S144). The reaction of *Z-6* with 1-butylamine to afford imine *Z-6A* in CDCl_3 (quantitative after 12 min) proceeded significantly faster than the corresponding reaction of *Z-3* (33% imine *Z-3A* after 65 h), showing the capability of chalcogen bonding for regulating dynamic covalent chemistry.

Responses to comments by Reviewer 2:

Comment: In this work the team of You describes the synthesis, experimental and theoretical analysis and application of a novel diarylethene-based photoswitch with four state behavior within a single switching unit. The diarylethene photoswitch is a special derivative as it is not ring-fused at the central double bond. This unlocks the option for additional cis-trans isomerization next to ring opening and closing upon light irradiation. However, it also causes problems because of the inherent difficulty to selectively address only one photo process at the time. The authors address this challenge by using deliberate and tunable chalcogen bonding via introduction of a tellurium-ether site and an aldehyde at the central double bond. In the neutral state of the tellurium-ether selective cis-trans photoisomerization takes place and the authors are able to obtain >90% switching to the *Z* isomer and up to 67% of the *E* isomer.

This selectivity changes when the tellurium-ether is methylated to the corresponding telluronium cation, which enhances the chalcogen bonding and abolishes cis-trans photoisomerization. Now electrocyclic ring closure and opening is observed with the corresponding strong color changes in very high overall yields. Methylation can be reversed by adding TBAI, which also renders the control over photochemical pathways reversible. The authors demonstrate this in a cyclic experiment cycling between essentially three different states of the system. They further provide an application for reversible writing and erasing of information on paper using photomasking. When reversible dynamic imine-chemistry is introduced, it is possible to bleach and recover the yellow color of the telluronium open *Z* isomer.

Overall, this study provides a very intriguing and well working concept for the establishment and control of multi-state photoswitching without resorting to oligomerization of switching units. The findings of this study represent a very high degree of conceptual novelty and performance and explore new and exciting possibilities for multi-addressability within small molecular building blocks. As stated above, gaining precision control over alternative types of photochemistry within the same molecular building block is highly challenging and I congratulate the authors to their very nice results. For these reasons I am happy to recommend publication of this work in *Nature Communications*, however a number of important points need to be addressed before, which are outlined below.

Response: Thanks a lot for the reviewer's comments and suggestions on our manuscript.

Major points:

Comment: One major point is the still lacking *Z* to *E* photoisomerization in almost all derivatives. Only for derivative **4** can more *E* isomer be accumulated than *Z*. This is of course a pity, as all the other three photo processes are much more effective. It is also curious as the photochromism should allow for very efficient *Z* to *E* photoisomerization with wavelengths >420 nm (e.g. Figure S48). I would strongly urge the authors to try 1. Longer wavelengths of light for this photoisomerization, different solvents with different properties, e.g. protic ones, very apolar ones like alkanes etc. It is also curious if not temperature effects play a major role here, and if not photoisomerizations at lower or higher temperatures are found more effective in this regard.

Response: Thanks a lot for raising the issue. We agree with the reviewer on the importance of enhancing *Z* to *E* photoisomerization and have made many efforts, including relevant experiments according to the reviewer's suggestions. For solvent effects, we conducted photoisomerization experiments for methanol, DMSO/H₂O, and *n*-hexane solutions of **3**, and found that the *Z*→*E* efficiency was not improved (Figures L1-L3). We also changed the wavelength of light (465 nm and 475 nm) and didn't get an effective boost (Figure L4). Furthermore, we tried to use 450 nm illumination for *Z*-**3** in DMSO-*d*₆ at 120 °C or *Z*-**3** in CDCl₃ at - 40 °C. The *Z*→*E* isomerization efficiency was improved at high temperature upon reaching the photostationary state (*E*:*Z* = 53:47) (Figure L5). The low temperature experiment in

CDCl₃ didn't improve the *Z*→*E* efficiency, yet requiring increasing time of illumination (Figure L6). The effects of acid on improving *Z*→*E* isomerization will be discussed later. Different from conventional *E/Z* photoswitches, for which *E*-isomer is generally thermodynamically favored, *Z*-isomer is more stable for **3-5** at room temperature, likely making a difference.

Fig. L1. (A) (Ai) ¹H NMR spectrum of *E*-**3** (10 mM) in CD₃OD at 25 °C; (Aii) Irradiation of *E*-**3** with UV light (365 nm, 25 min). The ratio of *E*:*Z* is 4:96; (Aiii) Further irradiation with visible light (450 nm, 50 min). The ratio of *E*:*Z* is 35:65. (B) UV-vis absorption spectra of *E*-**3** (60 μM in MeOH) in the pristine state and at the PSS upon irradiation with different wavelength of light (365 then 450 nm).

Fig. L2. (A) (Ai) ¹H NMR spectrum of *E*-**3** (10 mM) in DMSO-*d*₆/D₂O (8:2) at 25 °C; (Aii) Irradiation of *E*-**3** with UV light (365 nm, 25 min). The ratio of *E*:*Z* is 4:96; (Aiii) Further irradiation with visible light (450 nm, 50 min). The ratio of *E*:*Z* is 45:55. (B) UV-vis absorption spectra of *E*-**3** (60 μM in DMSO / H₂O) in the pristine state and at the PSS upon irradiation with different wavelength of light (365 then 450 nm).

Fig. L3. UV-vis absorption spectra of Z-3 (60 μM in *n*-hexane) in the pristine state and at the PSS upon irradiation with visible light (450 nm).

Fig. L4. (a) ^1H NMR spectrum of Z-3 (10 mM) in $\text{DMSO-}d_6$ at 25 $^\circ\text{C}$; (b) Irradiation of Z-3 with visible light (475 nm, 120 min). The ratio of *E*:*Z* is 13:87; (c) Irradiation of Z-3 with visible light (465 nm, 120 min). The ratio of *E*:*Z* is 45:55.

Fig. L5. (a) ¹H NMR spectrum of Z-3 (10 mM) in DMSO-*d*₆ at 25 °C; (b) Irradiation of Z-3 with visible light (450 nm, 60 min) at 120 °C. The ratio of *E*:*Z* is 53:47.

Fig. L6. (a) ¹H NMR spectrum of Z-3 (10 mM) in CDCl₃ at -40 °C; (b) Irradiation of Z-3 with visible light (450 nm, 180 min) at -40 °C. The ratio of *E*:*Z* is 38:62.

Comment: Another major point is the mentioned imine-chemistry. This is interesting as a color effect, but no details are given about the degree of imine-formation, the thermal stability of isomers or the photochemistry of the imine-products. Please provide more information in this regard and also explain more in detail in the text how this reversible functionalization affects the overall performance.

Response: Thanks a lot for the suggestions. We have performed imine formation reactions in solution, as Reviewer 1 pointed out. The reaction of Z-6 with 1-butylamine to create imine Z-6A (quantitative after 12 min) gave significantly faster

kinetics than that of **Z-3** (33% **Z-3A** after 65 h), showing the capability of chalcogen bonding for regulating dynamic covalent chemistry (Figures S135-S136, S139, and S142). The high yield formation of imine **Z-3A** was realized in the presence of molecular sieves. The absorption spectra of **Z-6** and **Z-6A** were measured to validate the color change (Figure S141). We have added the associated description and data in the main text and SI. We performed preliminary photoisomerization experiments for **Z-3A** and **Z-6A**, but the efficiency for the creation of **E-3A** and **C-6A** was low. The interplay between dynamic imine chemistry and photoswitches could open interesting prospects, and we will study it systematically and report the results in the future due to the length of the current manuscript.

Comment: What are the quantum yields for the different photo processes?

Response: We thank the reviewer for the suggestion. The quantum yields of representative compounds were measured and listed in Table S6.

Comment: The assignment of isomers is not always straight forward. I miss a thorough NMR analysis (NOEs, HSQC, HMBC etc.) for isomer identification.

Response: Thank the reviewer for the suggestion. We have performed additional NMR analysis, such as 1D ^1H - ^1H NOE and 2D ^1H - ^1H NOESY NMR spectra, to confirm the assignment of *Z/E* isomers (Figures S7-S8, S11-S12, S15-S16, S19-S20, S23-S24, S27-S28, S35-S36, S39-S40, S43-S44, and S47-S48).

Comment: Please add a table with all measured and calculated important data summarized. That should also include thermal parameters, which are not yet extracted from the heating experiments. Thermal data that need to be reported are the different relative isomer stabilities (ΔG s from the equilibrium isomer distribution reached after prolonged heating in the dark) and the associated first order kinetics (either with or without entering equilibria) and resulting Gibbs energies of activation ΔG^* and half-lives. Where this is not possible lower limits for these values should be given.

Response: Thanks a lot for the suggestions. The thermal stability of representative compounds was tracked through ^1H NMR analysis, and associated thermodynamic and kinetic parameters were calculated (Table S7).

Comment: Extinction coefficients are also missing for the photoswitches and their isomers.

Response: We thank the reviewer for raising the issue and added extinction coefficients in Tables S6.

Minor points:

Comment: The manuscript needs some English corrections, many sentences are hard to read and are not easy to understand.

Response: Thank the reviewer for the suggestions. We have made corrections in the revised manuscript to make sure our presentation as clear as possible.

Comment: Figure captions are hard to understand because letters were used in capital and small letters. Better use A for the image sections and then i, ii and so on for within the image sections. The small letters in the image sections are arranged somewhere in the text and it is not clear how much content belongs before or after them. Please always put them at the beginning, e.g. Ai) xxx Aii) etc.

Response: We thank the reviewer for raising the issue and have modified Figures in the main text according to the reviewer's advices.

Comment: Line 10: write: "and are challenging to control."

Response: Thanks a lot for the suggestion. We have made the correction.

Comment: All NMR spectra that are stacked, whether in SI or Paper Draft, should be reported in invert order. Because the first measured NMR is always listed as the final NMR and one must read from bottom to top.

Response: We thanks the reviewer for the suggestion and have modified Figures in the main text and SI according to the reviewer's advices.

Comment: The mixed nomenclature of *E/Z* and cis/trans needs to be unified, please decide for one of the two.

Response: Thank the reviewer for the suggestion. We have used the *E/Z* nomenclature in the main text and SI.

Comment: Naming of isomers and mechanisms is not well selected in general. Cycloreversion is abbreviated with *C* just as the closed form is abbreviated with *C*. *Z* also stands once for the *Z* isomer and once for photocyclization. It is best to use the term *E/Z* isomerization and label the isomers *Z/E* and photocyclization/cycloreversion is PC/CR and label the isomers o (open) and c (closed).

Response: Thank the reviewer for the suggestions. We agree the necessariness of choosing the appropriate abbreviation for photoisomers. The products of photocyclization and photocycloreversion are normally labelled as isomers c (closed) and o (open) in the literature. Since multistep *E/Z* switching and photocyclization/cycloreversion was discussed in this work, we simply used *C* (closed) for labelling photocyclization product as a way to maintain the same style as *E/Z* isomers. We added an explanation in the caption of Figure 1.

Comment: Line 10: write: "Here we demonstrate a multistep sequence of trans/cis..."; Line 13: "write: "...on open ethene bridged dithienylethenes offer a..."; Line 14: write: "While bidirectional *E/Z* photoswitching is dominated by neutral..."; Line 16: "Write: "the creation of cationic telluronium enables the..."; Line 17: "allows"; Line 37: Write: "Information encoding"; Line 38: type of blossoming not scientific; Line 39-40: "..., which can be utilized in..."; Line 45: "Adjacent chalcogen bonds" chalcogen bond.

Response: Thank the reviewer for the suggestions. We have made corrections in the revised manuscript.

Comment: Line 54, 55: Please define R₁ and R₂ in Fig. 1a.

Response: Thank the reviewer for the suggestion. The structures containing substituents R₁, R₂, and X in Fig. 1a were used to represent general DTE scaffolds with an open ethene bridge.

Comment: “chalcogen-containing dithienylethene photoswitches”. This is confusing naming, because the sulfur within the thiophenes of the parent dithienylethenes is also a chalcogen. This should be more clearly described.

Response: Thank the reviewer for the suggestion. We changed “chalcogen-containing dithienylethene photoswitches” to “chalcogen bond-containing dithienylethene photoswitches”.

Comment: Line 71: X: The X stands for chalcogen, which was previously abbreviated as Ch. This should be defined uniformly.

Response: Thank the reviewer for the suggestion. The “X” in this article (Figures 1 and 2) represents the substituent on the position 5 of thiophene.

Comment: Line 72: I would rather write a “Vilsmeier-Haack-Arnold formylation type” reaction, because it is not that classical.

Response: Thank the reviewer for the suggestion. We have made the change in the revised manuscript.

Comment: Fig 1: Please add all the stimuli at all arrows.

Response: Thank the reviewer for the suggestion. We have modified Fig. 1 in the main text according to the reviewer’s advices. Since light wavelengths for irradiation would be dependent on substituents R₁, R₂, and X, $h\nu_1$ and $h\nu_2$ were used for bidirectional photoswitching.

Comment: Fig 2: Please give more details about synthesis including reagents into the Figure; Fig 2: The reversibility at 2A (c) that you proclaimed in Figure 1 should be highlighted in this Figure too.

Response: Thank the reviewer for the suggestions. We have modified the Fig. 2A in the main text according to the reviewer’s advices.

Comment: Line 81: TeMe₂: TePh because it is Z-4.

Response: Thank the reviewer for the suggestion. In the sentence “Chalcogen bonding between TeMe₂ and formyl (Te ··· O 2.67 Å) also exists in telluronium Z-6, with DTE adopting an antiparallel configuration (Fig. 2B)”, we are describing Z-6, so TeMe₂ is used.

Comment: Line 89–92: Reaction condition should be with A in the reaction overview not with B.

Response: Thank the reviewer for the suggestion. We have included reactions conditions in Fig. 2A.

Comment: Page 6 line 102-114: It is a bit confusing, as there is a permanent change between NMR and UV-vis data.

Response: Thanks a lot for the reviewer's comments. We have modified Fig. 3B to include UV-vis spectra after $E \rightarrow Z$ isomerization and then $Z \rightarrow E$ isomerization to match NMR data.

Comment: Line 103-106: I am missing the temperatures you measured the NMRs when reaching the PSS.

Response: Thank the reviewer for the suggestion. We have added the temperatures for measuring NMR spectra in Figure captions. NMR spectra were recorded at 25 °C unless otherwise indicated.

Comment: Line 107: Why was the cycle experiment for compound **4** in DMSO not conducted as it is the best candidate for photoswitching?

Response: Thank the reviewer for the suggestion. We have added the photocycling experiment for **4** (Figure S67) according to the reviewer's advices. Due to minor side reactions upon prolonged irradiation of **4** the fatigue-resistance was modest.

Comment: Line 108: Please write "exhibiting reversibility and fatigue resistance over 12 consecutive irradiations with light of 365 nm and 450 nm."

Response: We thank the reviewer for the suggestion and made the change.

Comment: Please enter the NMR spectra in invert order.

Response: We thank the reviewer for the suggestion and made the change in Fig. 3A.

Comment: C: Len unit is missing, λ_{\max} is smaller than the rest;

Response: Thank the reviewer for the suggestions. We have modified Fig. 3C according to the reviewer's advices.

Comment: Line 108: 45% is not a great reversibility; Line 115: Te derivative (**3-5**) and S and Se derivative (**1, 2**); Line 128: write how many cycles instead of multiple.

Response: We thank the reviewer and have made the corrections. We have made attempts to improve the efficiency for Z to E photoisomerization. Please see the details in response to major point 1 described earlier.

Comment: Line 132-141: This section is not fully stringent in the argumentation. I think what happens after addition of acid is that the double character at the open bridge area is somewhat lost, and this is why the sterically more favoured *E* isomer is formed. Of course, this isomerization takes more time without adding acid. But it is a bit counter intuitive for me, that more polar solvents lead to greater thermal stability while addition of acid lowers it? How do you explain that?

Response: Thanks a lot for the reviewer's comments. We agree that the acid renders partial single bond character for the olefin bridge and accelerates *Z*→*E* isomerization. Without acid *Z*-**3** remained stable at room temperature in DMSO or chloroform, as well as at 80 °C in DMSO for 12 h. The acid promotes both *Z*→*E* isomerization and *E*→*Z* isomerization of **3** in CDCl₃ at room temperature, reaching the same equilibrium position after 5 min (60% *E*-**3**, Figure S96). The acid effect was also observed in DMSO-*d*₆ with a slower kinetics, and therefore, heating at 80 °C was needed (*Z*→*E* 550 min and *E*→*Z* 360 min) to give 58% *E*-**3** after reaching equilibrium (Figure S95). We clarified the description in the main text.

According to reference 55, we hypothesize that methanesulfonic acid (MA) would activate carbonyl group and reduce the energy barrier of isomerization (Figure L7). For solvent effects, we hypothesize that MA can directly activate the compound in chloroform. Differently, in competing solvents like DMSO the acid can bind to the carbonyl group and also DMSO (*J. Phys. Chem.* 1995, 99, 12214-12219), thus needing heating to promote the reaction.

Fig. L7. Possible mechanisms of *E/Z* isomerization.

Comment: Page 7 line 133 – 136: Compounds *Z*-**3**, *Z*-**4** and *Z*-**6** were heated to 80 °C in DMSO for 12 h only and there is no thermal isomerization happening. One should then test a higher temperature. By adding a very strong acid at 80 °C, 58% *E* is formed according to the text, but according to the NMR label only after 90 min at 80 °C? However, on page 8 line 141 it is then claimed that switching can be done with heat, light and acid. In fact, no evidence for thermal isomerization is given. -> Here, conclusions are drawn without providing more detailed evidence and, most importantly, acid and heat were used simultaneously and it is not clear which change is influenced by the heat and which by the acid.

Response: Thanks a lot for raising the issue. Please see the explanation in the previous response. We clarified the description in the main text.

Comment: Line 153: 425 nm is missing.

Response: We thank the reviewer and made the correction.

Comment: Line 157: fatigue resistance is shown, but for this rather low cycle number I would not say it is high.

Response: We thank the reviewer and deleted the word “high”.

Comment: Line 200–201: What is Z-6-I? Please define and then also name the molecule in Fig 5.

Response: Thank the reviewer for the suggestion. We have defined it in the caption of Fig. 5. The structure of Z-6-I is also include in Fig. 5B.

Comment: Fig. 5: please add the NMR data for the experiments shown in C for derivatives **5** and **7**, these data are crucial to evidence the claims.

Response: Thank the reviewer for the suggestions. We have modified Fig. 5C to show partial NMR spectra in the main text according to the reviewer’s advices.

Comment: Fig. 6 and imine chemistry-as stated above this part needs rework. It is not clear what imine chemistry is referred to and what is meant by “lock/unlock” in the Fig. 6. As it is, imine formation simply causes a color change without isomerization in the case of the telluronium form but not in the tellurium form. “Locking/unlocking” are therefore misplaced terms here that should be removed.

Response: Thank the reviewer for the suggestions. We performed imine reactions in solution, as detailed earlier. We modified Fig. 6 according to the reviewer’s advices.

Comment: Line 262: anti-counterfeiting; Line 273: 530-540 nm; Line 295: (C): C.

Response: We thank the reviewer and made the corrections.

SI:

Comment: From your table S5 it is not clear to me what you experimentally did. You started with the *E* isomers and enriched the *Z*-isomers, and then you irradiated the solution of the *Z*-enriched species to enrich the *E*-isomers? Please clarify the order of irradiation events and starting and end points.

Response: Thank the reviewer for the suggestions. The details of irradiation events were explained in note b below Tables S5 and S8.

Comment: In some cases, for example for compound **3**, the *E* and *Z* isomer assignments show some mistakes. It is written *E*-**3** but for, I guess, the *Z*-isomer is just written **3**.

Response: We thank the reviewer and made the changes.

Comment: It is not clear, how the *E* and the *Z* isomers were assigned, as ¹H NMR NOE experiments are missing. I guess NOE experiments were used to determine whenever it is the *Z* or the *E* isomer, but as it is crucial for this work it should be described in the SI.

Response: Thank the reviewer for the suggestions. As explained earlier, we have performed 1D ¹H-¹H NOE and 2D ¹H-¹H NOESY NMR analysis to confirm the assignment of *Z/E* isomers (Figures S7-S8, S11-S12, S15-S16, S19-S20, S23-S24, S27-S28, S35-S36, S39-S40, S43-S44, and S47-S48).

Comment: Do you mean *Z*-**4** for **4** at the NMR section on page S6?

Response: Thank the reviewer for the suggestion. Because of line breaking *Z*-**4** is not fully displayed. We have made corrections in SI.

Comment: Please add to the Figure captions of the NMR spectra shown in the SI what the measuring temperature was, and that you measured on 400 MHz and 141 MHz

Response: We thank the reviewer and made the changes.

Comment: Consistency: Isomeric ratio given sometimes 86/14 and sometimes 7:93 in the SI; Fig. S77 and S78 Sometimes 86/14 and sometimes 7:93.

Response: Thank the reviewer for the suggestions. The isomeric ratio with 86:14 is the isomer distribution of the compound **7** after synthesis, and 93:7 is the isomer distribution that reaches the PSS after illumination cycle of **7**. Therefore, the isomeric ratios are sometimes different. We adopted the same style to describe the ratio values.

Comment: Fig. S1-S30 Scale labeling with f1 (ppm) is not correct, measurement temperature and resonance frequency is not provided, sometimes spectra are not shown up to 0 ppm

Response: Thanks a lot for raising the issue. We have made corrections in supplementary information.

Comment: Fig. S44 Is this really an isosbestic point?

Response: Thanks a lot for the reviewer's comments. The side reactions with prolonged illumination time at low concentrations may complicate the spectra, although we have not observed this in NMR data.

Comment: Fig. S46-S47 What is the signal at 7.75 ppm? A side product?

Response: Thanks a lot for the reviewer's comments. The signal at 7.75 ppm is from a side product.

Comment: Fig. S50 Is this really an isosbestic point?

Response: Thanks a lot for the reviewer's comments. The side reactions with prolonged illumination time at low concentrations may complicate the spectra, as also observed in NMR data (Figure S67).

Comment: In S71 one can see a signal at 7.10 that is partially hidden with the figure on top.

Response: Thanks a lot for raising the issue. We have modified Figure S100 in supplementary information.

Comment: Fig. S64 B and c: What are the new signals? New species? Fig. S67 and S69 What are these new signals? Fig. S79 What are the new signals? Fig. S93 What are the new signals?

Response: Thank the reviewer for the suggestions. We measured ^1H NMR spectrum of methanesulfonic acid (MA) in CDCl_3 and confirmed that the new signals belonged to the acid (Figure L8). The formation of assemblies of MA via hydrogen bonds was reported (*Journal of Molecular Structure* 2005, 748, 77-90).

Fig. L8. ^1H NMR spectrum (400 MHz, 25 $^\circ\text{C}$) of methanesulfonic acid (MA) in CDCl_3 . The additional peaks are due to the formation of assemblies of MA via hydrogen bonding.

Comment: Fig. S66 and S68 This is not really a evidence that no thermal isomerization takes place

Response: Thanks a lot for the suggestions. As described earlier, the thermal stability of representative compounds was tracked.

Comment: Table S6 Traces of c, but *E* and *Z* together yield 100%?

Response: Thank the reviewer for the suggestion. For compound **5**, with the increase of illumination time we found that the color of the NMR solution was deepened due to the formation of a tiny amount of *C* isomer. We have added the ratio of *C* in Tables S6.

Comment: Fig. S77 and S78 Sometimes 86/14 and sometimes 7:93.

Response: We thank the reviewer and made the changes.

Responses to comments by Reviewer 3:

Comment: In the manuscript, the authors describe the synthesis and study of photoswitches that exhibit different switching properties depending on substituents. In the presence of a neutral tellurium substituent, *E/Z* photoisomerization is found. If the tellurium atom is converted to a cationic species, mainly a photocyclization/cycloreversion by irradiation takes place. As this conversion is reversible, the whole system represents a multistep photoswitch. The experiments are well thought out and well performed and the field itself is extremely exciting. Nevertheless, unfortunately, I cannot recommend acceptance for publication in Nature Communications due to the following reasons:

1) The first issue is the lack of novelty. It is already known that photoswitches can be affected by chalcogen bonds and thus one can regulate the switching ability (reference 40). In addition, it has already been shown that thiophenyl-substituted alkenes can be switched differently depending on the wavelength used (reference 23). The percentage of switching found there (Figure 5a) partly exceeds the distribution of isomers in the PSS described in this manuscript.

Response: Thanks a lot for the reviewer's comments and suggestions on our manuscript. The development of multistep photoswitching is very exciting, as the reviewer stated. With multiple species and processes involved, efficient control is a difficult task. We agree with the reviewer in the statement that photoswitches can be affected by chalcogen bonds and thus one can regulate the switching ability (references 39-42). However, these elegant example focuses on *Z/E* photoswitches, such as azobenzene and hemithioindio. The regulation of dithienylethenes (DTEs), one classes of the most employed phoswitches, by chalcogen bonds, hasn't been reported, to the best of our knowledge.

The current research results are novel for three points: 1. We demonstrate for the first time the manipulation of multistep photochromic switching of open-bridged DTEs through the manipulation of chalcogen bond interactions. The domination of *E/Z* photoisomerization and photocyclization/cycloreversion can be achieved with neutral tellurium and cationic telluronium, respectively. As Reviewer 2 pointed out, the

control of multistate photoswitching in single photochromic systems for alternative types of photochemistry is highly challenging. 2. The reversible nucleophilic substitution reactions of neutral tellurium and cationic telluronium allowed selective gating of photochromism on purpose and realization of multistate responses with multi-addressability, with the underlying mechanism elucidated. With reversible reactions on tellurium/telluronium underexplored, these findings enrich the toolbox of dynamic chemistry and materials. 3. By taking advantage of unique chalcogen bonding patterns dynamic imine chemistry is facilely modulated, enabling the construction of photo and pH-mediated multistate rewritable materials. The interplay between imine chemistry and photochemistry could open interesting prospects.

Furthermore, we have reviewed the literatures on thienyl-substituted alkenes, and the photocyclization ratios are listed: 60% (405 nm, *J. Org. Chem.* 1988, 53, 803-808), 65% (405 nm, *Org. Lett.* 2015, 17, 4802-4805), 34% (405 nm, *J. Org. Chem.* 2017, 82, 10960-10967), and 69% (334 nm, *J. Am. Chem. Soc.* 2021, 143, 9162-9168, reference 23). In this manuscript, we report photochromic DTEs that can be driven by visible light. The photocyclization ratios of **6** and **7** were 42% and 81% with 425 nm irradiation, respectively. The strategies and results reported herein would enable future applications in noncovalent interactions, molecular assemblies, photoswitches, surface engineering, and smart materials. Therefore, we feel that the current work has the novelty and significance required for publication in *Nature Communications*, as other two reviewers agreed.

Comment: 2) The second issue concerns the extent of switching. Switches **3** and **5**, which can also be switched as cations and thus represent the actual multistep photoswitches, show only minor switching behavior. In no case was an isomerization of more than 45% possible.

Response: Thanks a lot for the reviewer's comments. For switches **3** and **5**, their *E*→*Z* photoisomerization efficiency is high (95%), but the *Z*→*E* photoisomerization efficiency is modest. In order to improve *Z*→*E* switching efficiency, we have made many attempts, such as varying the light wavelength, changing the solvent, raising the temperature, and adding the acid, as detailed earlier. Taking **3** as an example, the *Z*→*E* photoisomerization efficiency can be increased to 60% by adding methanesulfonic acid (MA) (Figure S96). One reason for the modest *Z*→*E* efficiency is the thermodynamic preference of *Z* isomer due to chalcogen bonding. We will continue to study how to achieve efficient control of the different states of open-bridged photoswitches in the future.

Comment: 3) The third issue relates to the interpretation of the data. The authors explain the different behavior of the switches by the different chalcogen bond strengths. Switches **6** and **7** have very similar structures; specifically, the chalcogen bond strength should be almost identical for both. The NBO stabilization energies determined by the authors are 11.4 and 11.2 kcal/mol,

respectively. Nevertheless, the two systems show different switching behavior. For example, *E/Z* isomerization also takes place in compound **6**. However, this does not occur with compound **7**. If the strength of the chalcogen bond influences the switching behavior, these results are difficult to explain. The difference here probably stems from the different absorption behavior, so that the ratio of the isomers in the PSS is different for **6** and **7** (cf. reference 23)

Response: Thanks a lot for the reviewer's comments. The varying photoswitching patterns between neutral tellurium **3** and cationic telluronium **6** (or **5** and **7**) are interpreted according to the strengths and mechanisms of chalcogen bonding, and DFT calculations validated the explanation. We agree with the reviewer that the difference between cationic telluronium **6** (X = Me) and **7** (X = Ph) may be due to different absorption behaviors. The extended conjugation along with the presence of chalcogen bonding would contribute to the stability of *C-7* (fewer *E* isomer) and thus improved photocyclization efficiency of *Z-7*, as we described in the main text. Moreover, we changed the wavelength to carry out the irradiation experiment at 313 nm, and the isomer ratio was not much different from the data at 425 nm when the photostationary state was reached, with *Z-7* again giving enhanced photocyclization efficiency over *Z-6* (Figures L9 and L10).

Figure L9. (a) ¹H NMR spectrum of Z-6 (10 mM) in DMSO-*d*₆ at 25 °C; (b) Irradiation of Z-6 with UV light (313 nm, 150 min). The ratio of *E*:*Z*:*C* is 17:42:41; (c) Further irradiation with visible light (535 nm, 20 min). The ratio of *E*:*Z* is 17:83.

Figure L10. (a) ¹H NMR spectrum of **7** (Z:E = 86:14, 10 mM) in DMSO-*d*₆ at 25 °C; (b) Irradiation of **7** with UV light (313 nm, 360 min). The ratio of E:Z:C is 5:18:77; (c) Further irradiation with visible light (650 nm, 180 min). The ratio of E:Z is 5:95.

We believe that the revised manuscript is improved, and we hope that the manuscript is now acceptable for publication.

Thank you for your consideration of our work and best personal regards.

Sincerely,

Lei You

Lei You
 Professor
 Fujian Institute of Research on the Structure of Matter
 Chinese Academy of Sciences
 Fuzhou China, 350002

REVIEWERS' COMMENTS

Reviewer #1 (Remarks to the Author):

The authors have addressed all the reviewer's comments. I recommend this work for publication without further changes/corrections.

Reviewer #2 (Remarks to the Author):

In their revision You and coworkers provide a thoroughly reworked manuscript and SI and added a substantial amount of data, new experiments and explanations. I just name a few important ones, which the authors now deliver, i.e. extinction coefficients, quantum yields, NMR analysis of constitution and conformations, and scrutiny of the imine formation. Another major point I have raised was the insufficient Z to E photoisomerization in the neutral forms of the photoswitches. The authors have tried all the suggested experiments to improve the yield of the E isomer in the pss but sadly not successfully and the rather moderate E isomer accumulations remain. However, in DMSO solution at 120 °C it was possible to accumulate up to 53% of the E isomer and apparently in the presence of acid up to 60% can be accumulated for derivative 3, which makes it the dominant species, even if not by far. This is an interesting finding in itself and I would recommend the authors to include it not only in the answers to the Reviewer but also in the SI and mention it in the manuscript.

As requested by the editor, I have also looked over the responses to the comments from reviewer 3.

When taking into account what the authors deliver and what they have answered to mine and Reviewer 3's comments, requests and criticism I come to the following conclusion:

This work does indeed provide high novelty on the concept level because it describes a way to control multiple possible photoreactions within the open-ring diarylethene photoswitch framework via a highly interesting weak interaction, namely chalcogen bonding. Although chalcogen bonding has been invoked to control E/Z photoisomerizations, this is a different type of photoreactions where additionally electrocyclic reactions are invoked. Also to the best of my knowledge this has not been consciously been applied for the purpose of selecting such different photoreaction pathways. The authors agree that the differences between Me and Ph substituents on the Te-atom are possibly rather small with respect to chalcogen bonding but as Reviewer 3 suggested, the resulting photochromism changes are likely the reason. Overall it is well evidenced that it is indeed chalcogen bonding that is responsible for the observed behavior, not only because of the serial character changes going from S to Te, but also because of the substantial changes induced by the transition from neutral to cationic species. This interpretation is further supported by DFT calculations.

The authors further deliver very good performance on the cationic species 7 and with respect to the E to

Z photoisomerization. When invoking chemical stimuli like MeOTf, TBAI, or amine additions, multi-switching can be conducted with very good performance again and essentially three different states can be accumulated (former Figure 5c). The utility of this control is finally showcased by a light-mediated rewritable materials application.

For these reasons I support publication of this very exciting and comprehensive work in Nature Communications in the present form. I only would like to ask the authors to provide former Figure 5c again, as this showcases very nicely the achievable isomer accumulations and performance of the multi-switching in their system.

October 23, 2023

Dear Reviewers,

Thank you very much for reviewing manuscript NCOMMS-23-30805A. We have uploaded our revised manuscript entitled “**Selection of Isomerization Pathways of Multistep Photoswitches by Chalcogen Bonding**” that we wish to have considered for publication as an article in *Nature Communications*.

The reviewers provided very helpful comments, and we have made every effort to address the comments and improve this manuscript (the changes/additions are highlighted in red in a copy of the manuscript as Supporting Information). I have so indicated in my point-by-point response and revision summary, which follows.

Responses to comments by Reviewer 1:

Comment: The authors have addressed all the reviewer's comments. I recommend this work for publication without further changes/corrections.

Response: We thank the reviewer for the recommendation for the publication of this work.

Responses to comments by Reviewer 2:

Comment: In their revision You and coworkers provide a thoroughly reworked manuscript and SI and added a substantial amount of data, new experiments and explanations. I just name a few important ones, which the authors now deliver, i.e. extinction coefficients, quantum yields, NMR analysis of constitution and conformations, and scrutiny of the imine formation. Another major point I have raised was the insufficient *Z* to *E* photoisomerization in the neutral forms of the photoswitches. The authors have tried all the suggested experiments to improve the yield of the *E* isomer in the pss but sadly not successfully and the rather moderate *E* isomer accumulations remain. However, in DMSO solution at 120 °C it was possible to accumulate up to 53% of the *E* isomer and apparently in the presence of acid up to 60% can be accumulated for derivative 3, which makes it the dominant species, even if not by far. This is an interesting finding in itself and I would recommend the authors to include it not only in the answers to the Reviewer but also in the SI and mention it in the manuscript.

As requested by the editor, I have also looked over the responses to the comments from reviewer 3.

When taking into account what the authors deliver and what they have answered to mine and Reviewer 3's comments, requests and criticism I come to the following conclusion:

This work does indeed provide high novelty on the concept level because it describes a way to control multiple possible photoreactions within the open-ring diarylethene

photoswitch framework via a highly interesting weak interaction, namely chalcogen bonding. Although chalcogen bonding has been invoked to control *E/Z* photoisomerizations, this is a different type of photoreactions where additionally electrocyclic reactions are invoked. Also to the best of my knowledge this has not been consciously been applied for the purpose of selecting such different photoreaction pathways. The authors agree that the differences between Me and Ph substituents on the Te-atom are possibly rather small with respect to chalcogen bonding but as Reviewer 3 suggested, the resulting photochromism changes are likely the reason. Overall it is well evidenced that it is indeed chalcogen bonding that is responsible for the observed behavior, not only because of the serial character changes going from S to Te, but also because of the substantial changes induced by the transition from neutral to cationic species. This interpretation is further supported by DFT calculations.

The authors further deliver very good performance on the cationic species **7** and with respect to the *E* to *Z* photoisomerization. When invoking chemical stimuli like MeOTf, TBAI, or amine additions, multi-switching can be conducted with very good performance again and essentially three different states can be accumulated (former Figure 5C). The utility of this control is finally showcased by a light-mediated rewritable materials application.

For these reasons I support publication of this very exciting and comprehensive work in Nature Communications in the present form. I only would like to ask the authors to provide former Figure 5C again, as this showcases very nicely the achievable isomer accumulations and performance of the multi-switching in their system.

Response: We thank the reviewer for the recommendation for the publication of this work and the reviewer's comments and suggestions on our manuscript. We have added the relevant experiment data in Table S5 and modified Figure 5C in the main text according to the reviewer's advices.

We believe that the revised manuscript is improved, and we hope that the manuscript is now acceptable for publication.

Thank you for your consideration of our work and best personal regards.

Sincerely,

Lei You
Professor
Fujian Institute of Research on the Structure of Matter
Chinese Academy of Sciences
Fuzhou China, 350002